# Clinically Effective Molecules of Natural Origin for Obesity Prevention or Treatment

**DOI:** 10.3390/ijms25052671

**Published:** 2024-02-25

**Authors:** Gladys Maribel Hidalgo-Lozada, Angelica Villarruel-López, Karla Nuño, Abel García-García, Yaír Adonaí Sánchez-Nuño, César Octavio Ramos-García

**Affiliations:** 1Institute of Science and Technology for Health Innovation, Guadalajara 44770, Mexico; gladys.hidalgo@academicos.udg.mx (G.M.H.-L.); abel.garcia@academicos.udg.mx (A.G.-G.); 2Department of Pharmacobiology, University Center for Exact and Engineering Sciences, University of Guadalajara, Guadalajara 44430, Mexico; angelica.vlopez@academicos.udg.mx; 3Department of Psychology, Education and Health, ITESO Jesuit University of Guadalajara, Guadalajara 45604, Mexico; karlanuno@iteso.mx; 4Department of Medical Clinic, Health Sciences University Center, University of Guadalajara, Guadalajara 44340, Mexico; 5Division of Health Sciences, Tonalá University Center, University of Guadalajara, Tonalá 45425, Mexico

**Keywords:** natural compounds, functional foods, carotenoids, resveratrol, ellagic acid, berberine, probiotics, obesity, treatment, cardiovascular risk factors

## Abstract

The prevalence and incidence of obesity and the comorbidities linked to it are increasing worldwide. Current therapies for obesity and associated pathologies have proven to cause a broad number of adverse effects, and often, they are overpriced or not affordable for all patients. Among the alternatives currently available, natural bioactive compounds stand out. These are frequently contained in pharmaceutical presentations, nutraceutical products, supplements, or functional foods. The clinical evidence for these molecules is increasingly solid, among which epigallocatechin-3-gallate, ellagic acid, resveratrol, berberine, anthocyanins, probiotics, carotenoids, curcumin, silymarin, hydroxy citric acid, and α-lipoic acid stand out. The molecular mechanisms and signaling pathways of these molecules have been shown to interact with the endocrine, nervous, and gastroenteric systems. They can regulate the expression of multiple genes and proteins involved in starvation–satiety processes, activate the brown adipose tissue, decrease lipogenesis and inflammation, increase lipolysis, and improve insulin sensitivity. This review provides a comprehensive view of nature-based therapeutic options to address the increasing prevalence of obesity. It offers a valuable perspective for future research and subsequent clinical practice, addressing everything from the molecular, genetic, and physiological bases to the clinical study of bioactive compounds.

## 1. Introduction

Obesity is defined as an abnormal or excessive accumulation of fat that represents a health risk, and it is characterized by reaching a body mass index (BMI) equal to or greater than 30 kg/m^2^ [1]. The magnitude of this problem extends globally and poses a public health crisis. Its prevalence continues to increase in all age groups in Europe and some American countries [2], and it is estimated that more than 60% of the population is obese or overweight [1,2]. Ischemic heart disease, the main cause of death worldwide, is closely linked to obesity [3], which is also related to ischemic or hemorrhagic strokes [4], representing the second global cause of death, as well as high blood pressure, type 2 diabetes mellitus (DM2), metabolic syndrome, and cancer [3,5]. Furthermore, obesity is correlated with significant causes of work disability, such as low back pain and osteoarticular diseases, which significantly impact the financial resources of healthcare systems [5]. Despite this, the general population still fails to fully grasp the severity of this metabolic disorder [2].

Obesity has a complex origin, but diet is crucial to its development. In addition to the energy imbalance, some foods promote obesity, including simple carbohydrates such as sucrose (the main disaccharide of cane sugar), glucose and fructose (monosaccharides), highly hydrolyzed starches found in flour; saturated fatty acids, polyunsaturated fatty acids that promote the pro-inflammatory pathway of eicosanoids, such as arachidonic acid and linoleic acid; ultra-processed foods that contain a large number of additives, sweeteners, food colorings, preservatives, among others, related to obesity and other chronic non-communicable diseases. On the other hand, there are foods with molecules capable of regulating gene expression and metabolic pathways in a favorable way, known as phytochemicals, which are characterized by having a high antioxidant and anti-inflammatory activity. These phytochemicals are secondary metabolites of plants, vegetables and fruits such as those derived from phenolic compounds, derivatives of terpene compounds, some alkaloids, as well as some methylated purines (pseudoalkaloids). Likewise, plant-based primary metabolites are linked to health benefits, combating obesity. Some of these molecules deserve special attention since the treatment of obesity requires multiple interventions, not only with physical activity and nutritional therapy, but also with pharmacological therapy and/or surgery in specific cases. However, the currently available drugs have a high incidence of adverse effects and have variable effectiveness among patients. Many naturally occurring molecules have shown favorable effects in vitro or in vivo, but only some have been studied in humans. It is important to understand the mechanisms of action of the natural compounds or molecules that have shown clinical efficacy and analyze the results obtained in the clinical trials carried out to date.

For this review, the search for scientific articles was carried out in the search engines and electronic bookstores PubMed, Google Scholar, ScienceDirect, Dialnet, SciELO, Latindex, Redalyc, Medigraphic and Elicit using the keywords in English and Spanish: Obesity treatment, antiobesity molecules, weight loss natural molecules, obesity bioactive compounds, antiobesity drugs, antiobesity phytochemicals, antiobesity complementary medicines, antiobesity alternative medicines, obesity natural treatment, adipose natural molecules, obesity microbiome, obesity microbiota. Additionally, the obesity pathophysiology and the molecular and metabolic pathways described for each natural molecule were also reviewed. The selection of articles was carried out with the following criteria: (a) articles in English and Spanish published from 2000 to 2023, (b) original articles, narrative reviews, systematic reviews, and systematic reviews with meta-analyses written in English and Spanish, (c) theses and dissertations published in repositories of public and private universities in Spanish, (d) official websites of national and international organizations and public and private universities in Spanish and English, and (e) studies directly related to the search objective. Within the limitations of the review study, the following results were obtained: a reduced number of clinical trials carried out with natural molecules, most meta-analyses with the reviewed molecules used on preclinical studies, limited data from meta-analyses with these molecules in clinical studies, wide variability in the reports of clinical outcomes of the molecules of interest, and lack of information on adverse effects of several of the molecules analyzed. This review aims to analyze the clinical evidence and assess the efficacy and tolerability of key natural molecules in treating obesity.

## 2. Etiopathogenic Mechanisms of Obesity

Obesity is a chronic disease of multifactorial etiology that involves an energy imbalance, genetic and epigenetic factors, alterations in glucose and lipid metabolism, disorders of adipose tissue functioning, neuroendocrine dysregulation, and alterations in the intestinal microbiota, among others [6,7,8].

### 2.1. Genetic Factors

It is estimated that the genetic load may be responsible for the development of obesity by 40 to 70% [6]. So far, researchers have identified 227 gene variants and over 300 SNPs linked to obesity [9]. In genome-wide association studies, more than 1100 independent obesity-associated loci have been identified. Although monogenic obesity can happen, polygenic alterations are more frequently observed.

Some of the most studied alterations include the mutation of the leptin (*LEP*) and leptin receptor (*LEPR*) genes, which causes a deficiency of the LEP protein and, therefore, alterations in the satiety stimulus with increased appetite. Some rearrangements in the sequence of the proprotein convertase subtilisin/kexin type 1 (*PCSK1*), proopiomelanocortin (*POMC*), and melanocortin-4 receptor (*MC4R*) genes have also been studied, which cause the expression of the Agouti-related protein that, when binding to the melanocortin-2 receptor accessory protein (MRAP) 4, increases the appetite. Changes in the Ras2 suppressor protein kinase gene (KSR2) affect energy consumption and expenditure by interacting with adenosine monophosphate-activated protein kinase (AMPK) and the cyclic-adenosine monophosphate response-element-binding protein 3-regulatory factor (CREBRF), which participates in the storage and use of cellular energy [6].

Other widely studied alterations include loss-of-function variants of the gene that encodes adenylate cyclase 3, which is widely distributed, predominantly in adipose and subcutaneous tissue, but also participates in the correct functioning of the *MC4R* gene [10] and in the synthesis of cyclic adenosine monophosphate (cAMP), which functions as a second messenger in the glucagon-like peptide-1 (GLP-1) signaling pathway, the hormone ghrelin and the melanocortin-stimulating hormone [10], *POMC*, Fat Mass and Obesity (*FTO*) [11], interleukin 6 (*IL-6*) [12], perilipin (*PLIN*) [13], adiponectin (*ADIPOQ*) [14], *LEP* [15], *LEPR* [16], *POMC* [17] genes, the family of peroxisome proliferator-activated receptors (PPAR) [18] and the uncoupling proteins (UCPs) [19], which are related to the activation of thermogenesis. These genes and their metabolic and signaling pathways have a close relationship with systemic oxidative stress and inflammation, both critically related to the development of obesity and the appearance of many linked comorbidities [6,20].

### 2.2. Epigenetic Factors

Regardless of pre-existing genetic modifications, epigenetics plays a fundamental role in the expression of the obesity phenotype. Epigenetic modifications are modulations in gene expression without changes in the deoxyribonucleic acid (DNA) [21]. This modulation can be induced through DNA modifications, histone alterations, non-coding ribonucleic acids (RNAs), or adenosine triphosphate (ATP)-dependent chromatin remodeling complexes [22].

Histone methylation and DNA methylation are common modifications through which some environmental factors can prevent the expression of specific genes or DNA segments [20,21]. Histone methylation consists of adding a methyl group to lysine and arginine of histones, which are DNA packaging proteins. Methylated histones adhere strongly to the DNA, keeping the chain condensed, and, in this way, do not allow the integration of transcription factors, which means that the information from that section of DNA cannot be expressed. This process is also known as transcriptional gene silencing (TGS). DNA methylation occurs because DNA-methyltransferase adds a methyl group to cytokines, generally in regions rich in guanine and cytosine. These regions are called “CpG islands” and are part of the non-coding DNA, but their importance lies in the fact that they are usually located in the promoter region of most genes [22,23]. DNA methylation of CpG islands, in addition to maintaining the compact configuration of chromatin, prevents the integration of transcription factors and thus also produces gene silencing. Other modifications include acetylation, which favors gene expression; deacetylation, which inhibits gene expression; or modulation of the expression of some non-coding RNAs, such as microRNAs (miRNAs) [20,22,24].

The main importance of epigenetics is that unlike genotype and genetic changes, epigenetic changes are modifiable and, in many cases, reversible [24]. However, epigenetic modifications that persist can be replicated and conserved in mitosis or meiosis [22,24,25] so they can be transmitted to subsequent generations [26,27]. Epigenetic modifications can be induced by external factors such as diet, physical activity, inflammatory processes, stress, etc., and they can occur during pregnancy or throughout life [21,25], with critical moments of greater susceptibility in the fetal and neonatal stages. Although some prenatal epigenetic modifications are unstable, others remain in adulthood but can be corrected with interventions, such as exercise or favorable dietary changes [21,25,28].

Maternal malnutrition or obesity can cause intrauterine modifications that affect DNA methylation [21,26]. Similarly, obesity prior to pregnancy or overnutrition during pregnancy favors a pro-inflammatory state or insulin resistance that stimulates the release of pro-inflammatory molecules such as interleukins and tumor necrosis factor (TNF-α) and the expression of adipogenic factors in the mother and the fetus [26]. Exposure to these alterations can cause histone modification and DNA methylation, with a consequent increase in the expression of adipogenic and lipogenic genes that favor obesity in other stages of extrauterine life [26,27,29]. One of the genes studied is *POMC*, which, in cases of hypermethylation, can disrupt the regulation of food intake even in the presence of leptin and insulin [21]. Maternal obesity also promotes methylation and increases leptin expression, leading to hypersecretion and functional resistance to leptin from infancy or later in life [26]. Likewise, a high-fat diet during pregnancy produces more acetylation of histone H3K14 and lower sirtuin-1 (*SIRT1*) gene expression in the liver and heart. This decrease in *SIRT1* expression favors the presence of fatty liver, obesity, DM2, and diabetic cardiomyopathy [30,31].

The diet we have throughout our lives is crucial in epigenetics. Consuming high levels of saturated fat leads to DNA methylation in adipose tissue and alters the expression of 28 messenger RNAs. A diet with polyunsaturated fats induces multiple methylations, but these do not generate changes in gene expression, although both diets have a similar effect on weight gain [21]. High concentrations of palmitate increase histone acetyltransferase activity [27]. On the contrary, calorie restriction for more than eight weeks reduces hypermethylation in some genes affected by a high-fat diet. However, the adverse effects of insufficient overfeeding are quickly noticeable, while the positive effects of a calorie-restricted diet take longer to impact DNA methylation changes [21,27]. The above partly explains the benefits produced by metabolic surgery (restrictive, malabsorptive or combined technique) [32], which causes a sudden and sustained reduction in caloric intake, contributing to energy balance. After six to 12 months, it induces favorable effects on promoter methylation of the pyruvate dehydrogenase kinase 4 (*PDK4*) gene and proliferator-activated receptor coactivator g-1 alpha (*PGC-1a*), among other genes [21,33]. Dietary folate deficiency has been related to a more significant volume of fat mass, insulin resistance, increased risk of DM2, decreased DNA methylation of lymphocytes, and changes in DNA methylation of 236 CpG sites and genes associated with obesity [27].

A sedentary lifestyle, exercise, and moderate or intense physical activity have also been recognized as epigenetic modifiers. In individuals with a history of low physical activity, exercise for six months produced changes in the DNA methylation of 7663 genes, including some related to obesity, such as the cAMP response element-binding protein 4 (*CREB4*), elongation of very long-chain fatty acid proteins (*ELOV*), glucose transporter 4 (*GLUT4*), and hormone-sensitive lipase (*HSL*) [23,27]. Furthermore, a relationship has been reported between exercise and the expression of proinflammatory cytokines such as IL-6 and TNF-α [23,34]. Subjects who exercise periodically have lower expression of proinflammatory cytokines and are better adapted when they are released. Although exercise-induced methylation changes can be observed from the first session, the duration of this epigenetic modification is longer with chronic exercise exposure [23].

### 2.3. Energy Imbalance

Excessive caloric intake, compared to individual requirements, contributes to the development of obesity [7]. This explains the growing incidence of obesity, since food consumption high in fat and sugar has increased globally, coupled with decreased physical activity [7,35,36]. However, the fundamental cause of obesity is a state of positive energy balance characterized by caloric intake greater than energy expenditure. This originates from a complex interaction of behavioral, environmental, physiological, genetic, and social factors influenced by interactions between individuals, families, institutions, organizations, and communities [8].

A fundamental element in the regulation of energy balance is physical activity, which regularly requires 60 min per day for children and 150 min per week for adults [7]. Insufficient physical activity is favored by societal changes, leading to a significant decrease in total energy expenditure. Furthermore, when obesity occurs during childhood, it predisposes the individual to develop obesity in later stages of life [7].

### 2.4. Neuroendocrine Dysregulation

The regulation of energy expenditure and intake involves several body systems, such as the nervous system, digestive system (including the liver and pancreas), and adipose tissue cells, which play a crucial role [37]. Obesity is correlated with the dysregulation of many signaling pathways, including hormones, cytokines, neurotransmitters, neuropeptides, adipokines, chemokines, growth factors, proliferation molecules, differentiation factors, antioxidant enzymatic systems, non-enzymatic antioxidants, and the overexpression or underexpression of many genes [38]. These alterations can begin gradually and progressively, activating other neuroendocrine responses and promoting dysfunction. We summarize the principal occurrences of this pathophysiology below.

The foods consumed are digested into macronutrients (carbohydrates, lipids, and proteins) to be used in the body, and the excess nutrients are stored for later use. In the presence of insulin, excess of glucose is polymerized into glycogen in the liver and striated muscle. The excess of macronutrients can only be stored in adipose tissue, which requires being transformed into triglycerides in the liver. In its catabolic process, glucose generates acetyl coenzyme A (Acetyl-CoA) that can pass into the anabolic pathway for the synthesis of fatty acids (lipogenesis), mainly in adipose tissue and the liver [39]. Amino acids are catabolized to generate intermediates of the Krebs cycle, from where they can leave the mitochondria as citrate and be metabolized to form Acetyl CoA for lipogenesis [40,41]. These fatty acids are re-esterified and linked to glycerol-3-phosphate to form triglycerides, which are transported from the liver in very low-density lipoprotein (VLDL) molecules and stored in adipose tissue [42]. Mature adipocytes can store a large number of triglycerides, increasing their volume up to 20 times through hypertrophy [39]. Furthermore, this excessive growth favors adipocyte hypoxia, the release of reactive oxygen species (ROS), and inflammatory cytokines that activates adipogenesis (adipocyte hyperplasia), another mechanism to increase storage capacity [43].

Adipogenesis develops from multipotent mesenchymal stem cells related to preadipocytes, which remain available in the vascular stroma of adipose tissue and undergo mitotic clonal expansion and differentiation of adipocytes to white adipose tissue that has triglyceride storage functions, or to brown adipose tissue with thermogenesis functions [44]. The CCAAT/enhancer-binding protein β (C/EBP)β and C/EBP δ, which are highly sensitive to adipogenic stimuli, increase from the initial stages [43,45,46] and stimulate the expression of the C/EBPα, sterol regulatory element-binding binding protein 1 (SREBP-1) and the expression of the peroxisome proliferator-activated receptor γ (*PPARγ*), which has an essential role in the final phases of adipogenesis [46] and regulates the increase in adipogenic factors, including *C/EBPα* expression itself, in a positive feedback mechanism [43]. Because adipose tissue is not a passive storage space but rather a tissue with endocrine, paracrine, and autocrine functions, new mature adipocytes are capable of not only increasing storage capacity but also releasing adipocytokines, some of which function as neurotransmitters and participate in the neuroendocrine regulatory complex [47].

In healthy individuals, leptin, an adipocytokine, is typically present at 5 to 15 ng/mL [48]. However, its expression is influenced by various factors, including diet, insulin secretion, glucocorticoids, cytokines, and the amount of adipose tissue [48,49]. Under normal conditions, it can cross the blood–brain barrier by facilitated diffusion and induce satiety and energy consumption by stimulating POMC and inhibiting neuropeptide Y. However, high concentrations induce a state of leptin resistance that favors the presence of obesity and other metabolic alterations by limiting the stimulation of extracellular signal-regulated kinases (ERK) 1 and 2, mitogen-activated protein kinase (MAPK), and other downstream factors [50].

Adiponectin is secreted mainly by adipose tissue and has multiple functions, including (a) its participation as a promoter of the substrate of the insulin receptor IRS 1 and 2, which promotes insulin sensitivity; (b) activator of AMPK, which also promotes insulin sensitivity and the oxidation of fatty acids, and (c) MAPK activator, which stimulates glucose uptake [51,52]. However, obesity decreases serum adiponectin concentrations, with a consequent decrease in insulin sensitivity [14,51]. By reducing insulin sensitivity, insulin-dependent glucose transport decreases, which favors the presence of high plasma glucose concentrations and the consequent glucotoxicity that increases insulin resistance, creating positive feedback with glucose concentrations. Because insulin performs other functions, a series of metabolic pathways are triggered, including increased adipogenesis and lipogenesis [39].

### 2.5. Neurophysiological Factors

Multiple neurophysiological and psychological mechanisms related to adipose dysfunction or some food consumption participate in the regulation of appetite, hunger, satiety, and the psychological state, developing a bidirectional axis [53].

Many pathways are involved in this process, e.g., the dopaminergic system, that widely influences mood, addictions, motivation, and hunger regulation [54,55,56]. Dopamine acts on the nigrostriatal, mesolimbic, and mesocortical systems of the central nervous system by binding to the RD-1, 2, 3, and 4 receptors, resulting in appetite diminution. In obesity, these receptors decreased in number, affinity, and capacity to activate second messengers, thus inducing orexigenic effects [56].

The serotonergic system also plays an important role. Serotonin is produced from the amino acid L-tryptophan, present in some protein foods. In addition, sweet foods facilitate the crossing of L-tryptophan through the blood–brain barrier, as well as its 5-HT metabolite [57]. Various serotonin reuptake inhibitors have been used to treat obesity. The serotonergic system also plays an important role in regulating hunger, acting on the raphe nucleus and stimulating the neurotransmitter Gamma-Aminobutyric Acid (GABA) [57].

The GABA-mediated GABAergic system also can regulate satiety [58]. GABA is related to pleasurable sensations and the feeling of reward, like dopamine. There are two receptors for GABA, type A and type B, whose activation stimulates the Agouti-related peptide (AgRP), which, through negative feedback, inhibits GABA, generating greater food consumption. Subjects with depressive and anxiety disorders frequently present alterations in the regulation of this system [57,58].

The neural system also influences energy intake, which branches into two pathways, cannabinoids and opioids. This system controls enkephalins, nociceptins, and endorphins, which influence hunger and the consumption of palatable foods [58]. Obesity patients regularly have a deregulation of this neurophysiological system, thus increasing their energy intake. The receptors involved in this system are R-CB1 and R-CB2 in the cannabinoid pathway, and R-δ, R-κ, and R-µ in the opioid pathway [58]. Stress represents another important factor in the etiology of obesity [54,55]. Acute stress inhibits appetite; however, under continued stress, adrenal glands release cortisol, which increases appetite, particularly for foods high in fat, sugar, or both, with inhibition of the limbic system [54].

Stress is also related to sleep deprivation, which contributes to weight gain [54]. The “emotional ingestion” is another risk factor for obesity. It is characterized by food intake as a way to suppress or attenuate negative emotions [53,54]. All of these feelings, emotions, and psychological states interact directly with neurophysiological systems and food, forming a multidirectional axis of emotions–neurotransmitters–food where each of these variables influences the others.

### 2.6. Gut Microbiota

Another important factor in the etiopathogenesis of obesity is the intestinal microbiota. This is the set of communities of living microorganisms colonizing the intestine and plays a critical role in the health–disease balance in the individual. The composition of the human microbiome results from millions of years of coevolution and selective pressure, selecting a specialized community to live in the intestinal environment and achieving a mutualistic relationship and a state of balance with the host that is beneficial, both for humans and for the microorganisms they harbor [59]. Among all of the ecological niches found in the human body, the gastrointestinal tract (GIT) is the one with the greatest diversity and abundance of microbial taxa. In the proximal regions of the GIT, there is a low concentration of microorganisms due to the acidic pH and the rapid transit of the bolus; meanwhile, in the distal region (colon), the most significant number of microbes is found [60].

The intestinal microbiota has a vital role in energy balance since these microorganisms can metabolize an enormous amount of compounds from the diet, produce metabolites related to the hunger–satiety process, as well as a large number of hormones and neurotransmitters such as serotonin, which induces satiety [61,62,63]. In humans, the gut microbiota mainly comprises five distinct phyla: *Actinobacteria*, *Bacteroidetes*, *Verrucomicrobia*, *Firmicutes*, and *Proteobacteria*. *Bacteroidetes* and *Firmicutes* represent up to 90% of all intestinal bacteria [64]. The microbiota’s impact on host physiology is supported by a growing body of evidence that highlights multiple pathways, including improved energy utilization, immune system changes, metabolic signaling, and inflammatory pathways [59].

Dysbiosis, an alteration in the composition of the microbiota, has been linked to three distinct mechanisms that can occur at the same time: (a) the loss of “beneficial” bacteria, (b) an overgrowth of potentially dangerous bacteria, and (c) a reduction in microbial diversity [65]. It can also lead to an increase in caloric intake to the detriment of basal and total energy expenditure, but a correlation has also been identified between the existing intestinal microbiota and metabolic and nutritional status. In this way, healthy individuals have a greater amount of *Bacteroidetes* than *Firmicutes*, but industrialized diets low in dietary fiber cause a decrease in *Bacteroidetes* and an increase in *Firmicutes* [63,65]. In addition to bacteria, intestinal archaea, fungi, and viruses participate in the etiopathogenesis of obesity [63].

The microbiota can contribute significantly to the promotion of weight gain and fat storage and the generation of insulin resistance [66]. DM2, a pathology associated with obesity, also involves dysbiosis, which can also favor infections due to changes in the regulation of receptors in immune cells. An example is infection by fungal species of the genus *Rhizopus*, which is the causal agent of the severe infection called rhinocerebral mucormycosis [67]. The correlation between obesity and the bacterial composition of the human microbiome has been established through previous investigations, with particular attention given to members of the *Christensenellaceae* family and the *Methanobacteriales*, *Lactobacillus*, Bifidobacteria, and *Akkermansia* genera [68].

More recently, it has been identified that the *Christensenellaceae* family is linked to weight loss, showing an inverse correlation between its relative presence and an individual’s body mass index (BMI) [69]. In particular, it has been shown that *Akkermansia muciniphila* plays a crucial role in regulating body weight, and supplementation with this bacterium significantly improves the metabolic parameters of overweight and obese individuals [70].

The *Lactobacillus* and *Bifidobacterium* genera are recognized for promoting intestinal health. They have been used as traditional probiotics that significantly affect the balance of the intestinal microecology in humans, but the effects on body weight in overweight individuals vary depending on the species of *Lactobacillus*; specifically, a negative correlation was found between the abundance of *Lactobacillus paracasei* and obesity, while the abundance of *Lactobacillus reuteri* and *Lactobacillus gasseri* showed significant correlations with obesity development [71]. Therefore, we can summarize that the participation of the intestinal microbiota in weight control focuses on bacterial diversity, some specific species found, the metabolites that bacteria produce and are absorbed intestinally, the relationship between the microbiota, the immune system, inflammation, and oxidative stress, the participation of metabolites in gene expression, acting as transcription factors, and the capacity of the microbiota to degrade, biotransform and metabolize compounds from the diet [72].

### 2.7. Other Etiopathogenic Factors

There are medications associated with the development of obesity, known as obesogenic medications, which include mineralocorticoids, glucocorticoids, some antidiabetic drugs, insulin, and some antihypertensive medications, among many others. On the other hand, there are obesogenic molecules known as endocrine disruptors, mainly in foods (due to agrochemicals) and plastic containers, which interfere with the function and/or signaling of various hormones, neurotransmitters and neuropeptides, including androgenic and progestin sex hormones and peptides related to satiety and hunger, such as LEP, ghrelin, peptide YY, neuropeptide Y, GLP-1, and serotonin [73].

Circadian rhythms related to sleep–wake also play an important role in controlling energy balance, whose deregulation can cause greater hunger and lower basal energy expenditure. Several mediators regulate this system; however, the most important are cortisol, melatonin, growth hormone, and *CLOCK* genes [74]. There are viruses related to the obesity process, known as obesogenic viruses, among which is *Adenovirus* 36, which increases insulin sensitivity in adipocytes, generating more significant lipogenesis while reducing the expression and secretion of leptin, thus increasing hunger [75]. Likewise, many psychological, cultural, social, economic, climatic, political, and geographical factors determine eating patterns, the type of food available, the amount of physical exercise performed, as well as the quantity and quality of the food consumed [76].

## 3. Molecules with Demonstrated Clinical Efficacy in the Prevention or Treatment of Obesity

Obesity requires a comprehensive approach to treatment that includes pharmacological interventions in many cases. The drugs currently available offer variable results and a high incidence of adverse effects, which motivates research into new molecules. Clinical trials carried out to date have shown that some molecules of natural origin can achieve favorable modifications for the management of obesity.

### 3.1. Epigallocatechin-3-Gallate

Epigallocatechin-3-gallate (EGCG) is a polyphenol flavone-3-ol, catechin ester of epigallocatechin and gallic acid with three aromatic rings linked by a pyran ring, contributing to its functional benefits [77]. It is primarily found in green, red, white, black, and oolong teas derived from the tea plant (*Camellia sinensis*). Oral administration of EGCG has been shown to have high bioavailability. It is mainly absorbed in the small intestine, while the intestinal microbiota plays a fundamental role in its absorption in the colon [78]. It is suggested that EGCG is transported to the liver or metabolized by the gut microbiota to other potential metabolites because EGCG can be detected in plasma after ingestion, and its metabolites can be identified in bile, plasma, and urine [79].

EGCG exerts its effects mainly by regulating the suppression of hydrogen peroxide and ROS metabolism, the suppression of oxidative stress, the activation of fatty acid transport, and cholesterol oxidation and metabolism. Likewise, EGCG has been shown to significantly increase the expression of *FOXO1*, *Sirt1*, *CAT*, *FABP1*, *GSTA2*, *ACSL1*, and CPT2 proteins while decreasing the levels of nuclear factor kappa β (NF-κB), ACC1, and FAS proteins in the liver [80].

Studies found that EGCG prolonged lifespan by improving free fatty acid metabolism and reducing inflammatory and oxidative stress levels in preclinical models [80]. EGCG also activates the phosphatidylinositol 3-kinase (PI3K)/protein kinase B (AKT) molecular pathway, thus increasing the translocation of glucose into the cell and its utilization in cytoplasmic microtubules, improving glycolysis and energy metabolism [81]. EGCG has also been shown to significantly decrease levels of total free fatty acids (FFA), saturated fatty acids (SFA), and the n − 6/n − 3 ratio, as well as significantly increasing n − 3 FFA related to longevity and regulating inflammation (Table 1) [82].

EGCG has been shown to intervene in the regulation of various metabolic and molecular pathways involved in obesity, reducing glucose, serum lipids, inflammation, oxidative stress, and blood pressure, which are also involved in premature aging [83]. The results of clinical studies suggest that EGCG can prevent increases in body weight, BMI, and visceral fat, mainly in individuals under 50 years of age [82]. However, it does not produce a decrease in these variables [82,83]. Table 1 shows the clinical effects of epigallocatechin-3-gallate on obesity and related comorbidities, according to clinical trials and systematic reviews with meta-analyses of clinical trials found in the scientific literature.

### 3.2. Ellagic Acid

Ellagic acid is a polyphenol from the ellagitannin group; it is a derivative of chromene-dione and is a dimer of gallic acid. This compound exhibits a hydrophilic group consisting of four hydroxyl groups, two lactones and a lipophilic group incorporating two hydrocarbon rings [84]. These characteristics give ellagic acid the ability to act as an electron acceptor in various substrates and to participate in redox reactions [85]. Ellagic acid is found in a wide variety of fruits (pomegranates, persimmons, raspberries, strawberries, peaches, plums, kiwis), oilseeds (walnuts, almonds), and vegetables. It can be present as a free form or as derivatives that can be hydrolyzed under physiological pH and by the intestinal microbiota, thus increasing plasma levels of ellagic acid after ingestion of fruits and nuts [84].

Ellagic acid exerts antitumor, antioxidant, anti-inflammatory, antimutagenic, antibacterial, and antiallergic effects [86]. Its multiple pharmacological properties act on various signaling pathways, including the VEGFR-2, Notch, PKC, COX-2, PI3K/Akt, JNK (cJun) signaling pathway, and the mitochondrial pathways Bcl-2/Bax, TGF-β/Smad3, MMP SDF1α/CXCR4, MAPK, PDK-1, mTOR, p-ERK, NRF2, and p-JNK. Ellagic acid was also found to reduce the expression of histone deacetylases (HDAC), histone deacetylases to inhibit neovascularization, a common process in cancer and obesity. By analyzing the mechanism of action, it was revealed that ellagic acid inhibits hypoxia-induced angiogenesis by suppressing HDAC-6 in ECV304 cells. Furthermore, the knockdown of endogenous HDAC6 via small interfering RNA abolished hypoxia-induced expression of *HIF-1α* and *VEGF* and blocked the Akt activation. This interaction explains the effect of ellagic acid in modulating inflammation and oxidative stress [87,88].

Ellagic acid inhibits adipogenesis in two phases: adipocyte differentiation and clonal expansion [89,90]. This effect is because it produces a downregulation of adipogenic factors such as C/EBP alpha, Krox20, KLF4, and KLF 5, and inhibits the expression of *PPAR* gamma and aP2 by decreasing the expression of mRNAs for these proteins, which varies with the concentration of ellagic acid [89]. Furthermore, in the presence of ellagic acid, the mRNA expression of the preadipocyte secretory factor (Pref), called Pref-1 or Dlk1/FA1, increases, which prevents adipogenesis by activating the ERK/MAPK pathway [89,91]. These regulations produce inhibition of cyclin A expression and phosphorylation of retinoblastoma protein, key regulators of the cell cycle, which stops the progression between the G0/G1 and G1/S phases of preadipocytes found in these early phases, preventing their conversion to mature adipocytes [89,90]. Ellagic acid also interferes with triglyceride synthesis by decreasing the protein Lipin-1 and diacylglycerol acyltransferase-1 expression, which catalyzes the final acylation step in the triglyceride synthesis pathway [89]. This decrease may contribute to the prevention of hyperplasia of adipose tissue.

In clinical studies, ellagic acid has shown effectiveness in reducing waist circumference [92,93,94], which represents a decrease in central adiposity due to the correlation between them [95]. The reduction ranged between −0.7 and −3.8 cm. This effect is especially relevant due to the close relationship between abdominal fat and the risk of fatal and non-fatal cardiovascular disease [96,97]. Although the impact of ellagic acid on the percentage of body fat in humans has been reported [93], some authors suggest that ellagic acid causes a redistribution of body fat [98] and an increase in brown fat with a decrease in white adipose tissue [99], which can keep the amount of total body fat mass constant. To date, no adverse effects of ellagic acid have been reported, and it has a wide therapeutic margin [92,100]. However, the optimal dose still needs to be defined and subjected to evaluation in phase 3 clinical studies.

Table 2 shows the clinical effects of ellagic acid on obesity and related comorbidities, according to clinical trials and systematic reviews with meta-analyses of clinical trials found in the scientific literature.

### 3.3. Resveratrol

Resveratrol (3,5,4′-trihydroxy-trans-stilbene) is a molecule belonging to phenolic compounds that acts as phytoalexin in various plants and fruits. Specifically, it is classified within the group of stilbenes (existing in cis and trans configurations). It is found in a great diversity of plants and fruits; the best-known and most used source is the red grape (*Vitis vinifera*) [101].

Resveratrol can induce the expression of glutathione-S-transferase (*GST*) and glucuronyltransferase, both belonging to second-phase hepatic enzymes, and it inhibits several subfamilies of the cytochrome P450 (CYP), including the CYP1A1 isoenzyme [102]. This comes with important changes in the metabolism of molecules related to cancer, inflammation, and oxidative stress due to the increase in detoxification mediated by conjugation, unlike first-phase hepatic metabolism, which is mediated by redox reactions. Resveratrol also has immunoregulatory properties mediated by decreasing the expression and activation of the TLR4 receptor in monocytes and macrophages, thereby reducing the activation of NF-κB and IL-1 beta, as well as inhibiting the proinflammatory JAK/STAT3 pathway and matrix metalloproteinases 3 and 9 [103]. Resveratrol also inhibits the expression of *COX2*, thereby reducing the synthesis of proinflammatory prostaglandins of the E2 series [104].

About 70% of dietary resveratrol is absorbed after ingestion. However, only 0.5% is ingested, and resveratrol becomes systemically bioavailable [105]. Resveratrol is perceived as a xenobiotic within the intestine. It passes to the enterocytes lining the small intestine, where it is metabolized (including conjugation with sulfate and glucuronate) with the generation of polar metabolites (to optimize excretion) [101]. Conjugated resveratrol is transported to the blood from the enterocytes by binding to an ATP-binding cassette transporter called multidrug resistance protein 3 (MRP3). It is worth mentioning that the enzymes responsible for this Phase II metabolism (including sulfotransferases and glucuronosyltransferases) can manifest genetic polymorphisms that probably underlie some of the interindividual variability in the biological effects of resveratrol and its metabolites [106].

Current preclinical and clinical evidence on the positive effects of resveratrol in the treatment of obesity and related comorbidities is linked mainly to the regulation of various metabolic pathways and signal transduction, with resveratrol acting as a transcription factor, as well as an estrogenic hormone in some cases [107]. The main mechanisms of action of resveratrol are associated with its anti-inflammatory and antioxidant effects, helping to resolve inflammation and activate various endogenous antioxidant enzyme systems such as Nrf2 and ARE (antioxidant response element) for the synthesis of antioxidant enzymes such as superoxide dismutase (SOD), glutathione peroxidase (GSH-Px) and catalase (CAT) [108]. In vitro and in vivo studies have shown that different doses of resveratrol exert anti-obesity effects on 3T3-L1 adipocytes through various mechanisms, such as induction of apoptosis, decreased fat accumulation and adipogenesis, inhibition of cell differentiation through the activation of AMPK, and the induction of SIRT1-dependent cell apoptosis [109]. In randomized controlled clinical trials, resveratrol has shown significant reductions in weight and BMI compared to the placebo group (*p* < 0.05) [110,111,112,113,114,115]. Table 3 shows the clinical effects of resveratrol on obesity and related comorbidities, according to clinical trials and systematic reviews with meta-analyses of clinical trials found in the scientific literature.

### 3.4. Berberine

Berberine is an alkaloid present in plants such as *Berberis vulgaris*, *Berberis aristata*, *Mahonia aquifolium*, *Hydrastis canadensis*, *Xanthorhiza simplicissima*, *Phellodendron amurense* and *Rhizoma coptidis* [116]. It has been studied for its effects as a lipid-lowering, hypoglycemic, antibiotic, antioxidant, potassium channel blocker, and antineoplastic agent [117]. In obesity studies, berberine decreased adipocyte differentiation through downregulation of *C/EBPα* and *PPARγ* mRNA expression in visceral fat [118]. Furthermore, it upregulates the mRNA expression of the transcription factors *GATA-2* and *GATA-3* [119]. These factors control the transition from preadipocyte to adipocyte [120], participate in differentiation to brown adipose tissue, and suppress PPARγ, the basal activity [121].

Berberine also causes deacetylation and stimulation of AMPK expression in myotubes and adipocytes [121,122]. It also activates thermogenesis by activating the AMPK/SIRT1 signaling pathway [123]. On the other hand, it decreases insulin resistance by inhibiting the expression of *C/EBPα* and the phosphorylation of IKB kinase beta (IKKbeta) Ser (181) and IRS-1 Ser (307) [124,125,126]. It has been observed that berberine also reduces the expression of the genes *LXR*, *PPAR*, *SREBP-1c*, fatty acid synthase, acetyl-CoA carboxylase, acyl-CoA synthase, and lipoprotein lipase [121,127]. Additionally, other mechanisms associated with berberine have been identified, including its ability to inhibit hepatic gluconeogenesis through the phosphoenolpyruvate carboxykinase (PEPCK), glucose-6-phosphatase (G6Pase), and AMPK enzymes [128]. Many of the effects of berberine in obesity are related to its activity as an AMPK agonist, which also favors GLUT4 translocation and reduces glucotoxicity and insulin resistance, and, therefore, reduces the lipogenic stimulus.

Further, berberine inhibits the activation of macrophages by oxidized low-density lipoproteins (LDL) and the formation of foam cells by improving the expression and translocation of *LXRα-ABCA1* and, thus, cholesterol efflux [127,128]. In addition, it modifies the intestinal microbiota, improving the Firmicutes/Bacteroidetes ratio, regulating the production of microbiota metabolites, and consequently causing weight loss and improvement in the lipid profile [129].

In human studies, berberine has been found to have a beneficial impact on gene regulation related to cholesterol absorption when administered at a daily dose of 300 mg and has improved glucose homeostasis with a daily dose of 1 g [128]. Multiple studies in overweight or obese patients have reported the effects of berberine in reducing waist circumference and waist/hip ratio [130,131]. Some studies have also reported a decrease in body mass index, but the results on body weight have not been significant in most studies [130,132]. Recent studies report a decrease in visceral fat [133] in overweight patients, in addition to other metabolic effects of berberine, such as an improvement in the glycemic profile and a decrease in insulin resistance, which make it interesting as an auxiliary in the treatment of obesity [125,133].

Although some clinical studies carried out with berberine have reported adverse effects such as constipation, diarrhea, nausea, or abdominal discomfort [134,135,136], to date, no serious adverse effects have been reported. There are toxicity studies [137] that report a wide range of adverse effects and a wide therapeutic margin, so it is considered safe and well tolerated [133,138]. More studies are required to analyze whether its effectiveness may be greater in certain specific groups and determine its uses. Table 4 shows the clinical effects of berberine on obesity and related comorbidities, according to clinical trials and systematic reviews with meta-analyses of clinical trials found in the scientific literature.

### 3.5. Anthocyanins

Anthocyanins are flavonoid pigments of plant origin responsible for the colors of various fruits, flowers, and leaves. Anthocyanins range from orange-red to bright red, violet, and blue. Anthocyanins are water-soluble flavylium salts structurally composed by coupling a sugar unit to an anthocyanidin [139]. There are more than 600 types of anthocyanins, the main differences between them being the position and number of hydroxyl groups, the degree of methylation of the hydroxyl groups, the nature and number of sugar molecules present, and the acids attached to the sugars. Despite the large number of anthocyanin molecules already cataloged in the food matrix, only an-thocyanins derived from six types of anthocyanidins: pelargonidin, cyanidin, peonidin, delphinidin, petunidin and malvidin, have been extensively researched. [139,140].

The mechanism of action of anthocyanins is linked to their reducing capacity and their potential to act as a positive and negative transcription factor for many genes related to inflammation/anti-inflammation and oxidation–reduction. Cyanidin-3-glucoside, delphinidin-3-glucoside, and petunidin-3-glucoside can inhibit NF-κB activities through mitogen-activated protein kinase (MAPK) pathways. In contrast, cyanidins inhibit NF-κB activities of the enzyme cyclooxygenase type 2, involved in the production of proinflammatory series prostaglandins (E2 series) [141]. Furthermore, cyanidin-3-glucoside has reduced lung inflammation in rats [142]; in the same way, cyanidins have shown effects in reducing TNF-α, IL-6, MCP-1, iNOS, IL-10, serum glucose, total cholesterol, LDL, and C-reactive protein (CRP) [143]. Regarding their bioavailability, anthocyanins have very low absorption, bioavailability, and systemic distribution. However, the bioavailability of anthocyanin metabolites is 42 times greater than that of the original anthocyanins [144]. The identified anthocyanin metabolites have shown to improve human health [141,145].

Many preclinical and clinical studies have investigated the positive influence of anthocyanins on weight, obesity, and inflammation. In in vitro studies with cell cultures, strawberries had a notable anthocyanin richness and exhibited a high antioxidant capacity. One study found that strawberries, as opposed to bananas and oranges, induced a significant reduction in oxidative stress-induced neurotoxicity in PC12 cells previously exposed to hydrogen peroxide (H_2_O_2_) [146,147].

In a 14-week preclinical study with six-week-old male C57BL/6J mice using a dose of murray extract (*Lycium ruthenicum*) high in various anthocyanins at a 0.8% concentration dissolved in drinking water (orally), the extract increased the diversity of cecal bacterial communities, decreasing the *Firmicutes*/*Bacteroidetes* ratio, which has been related to lower weight gain and lower body mass index (BMI). Likewise, the species of the *Akkermansia* genus increased, and those of the *Faecalibaculum* genus decreased, resulting in a decrease in body mass [148]. Another 12-week study in nine-week-old male Wistar rats using raspberry extract and fructooligosaccharides (FOS) orally at a dose of 0.64% and 3%, respectively, of the diet, with raspberries being high in anthocyanins, demonstrated an increase in bacterial species, genera and families in the intestinal microbiota of rats related to an increase in the production of short-chain fatty acids (SCFAs), which have been involved in improvements in energy metabolism and body weight [149].

In animal models, purple dye from potatoes demonstrated a protective capacity against the generation of ROS and restored glutathione levels in mice subjected to a high-fat diet [150]. Also, oral supplementation with anthocyanins from blackberry and blueberry mitigated oxidative stress and inflammation through an increase in the levels of first-line antioxidant enzymes, such as superoxide dismutase (SOD) and glutathione peroxidase (GPx). Likewise, its administration over 12 weeks prevented weight gain in C57BL/6 mice with obesity induced by a high-fat diet [151]. Meanwhile, anthocyanins derived from mulberry and cherry reduced body weight and improved SOD and GPx activities in mice fed with a high-fat diet [147,152].

In clinical research, healthy individuals who incorporated 500 g of strawberries, rich in anthocyanins, daily for a month experienced a reduction in their risk of developing cardiovascular disease, evidenced by improvements in lipid profiles and antioxidant capacity [153]. In another trial carried out in participants with DM2 who received oral supplements of purified anthocyanins for 24 weeks, improved lipid profiles, antioxidant capacity, and insulin sensitivity were observed due to oral administration [147,154].

Table 5 shows the clinical effects of anthocyanins on obesity and related comorbidities, according to clinical trials and systematic reviews with meta-analyses of clinical trials found in the scientific literature.

### 3.6. Probiotics

Probiotics are live microorganisms (such as bacteria and yeast) that provide health benefits when consumed. They are naturally present in some fermented foods, added to food products, and available as dietary supplements [157]. In the market for food, supplements, and medicines, there are a wide range of probiotics that must be categorized at the strain level due to the varying effects they have on human physiology [158]. Most probiotic bacteria belong to the genera *Lactobacillus* (recently, many of them were renamed into other genera according to new phylogenetic classifications) and *Bifidobacterium*, which are Gram-positive. However, there are also Gram-negative bacteria with probiotic benefits, such as *Escherichia coli* Nissle 1917, *Akkermansia muciniphila*, and *Christensenella minuta*, among others, many of which are under study since they are considered “second generation probiotics” [159]. Various yeasts are used as probiotics, among them *Saccharomyces boulardii* and *Saccharomyces cerevisiae*. Various probiotics belonging to the genera mentioned above have been shown to positively affect weight control, obesity, and associated comorbidities [160].

In clinical trials, the second-generation probiotic *Akkermansia muciniphila* improves insulin sensitivity and significantly reduces insulinemia, serum total cholesterol, total body weight, and fat mass [161]. Meanwhile, *Lactobacillus gasseri* has some interesting effects on obesity [162] since it is a habitual resident of the human intestine and can modulate the intestinal niche. Its presence confers beneficial effects related to biliary tolerance and the deconjugation of bile acids, improving hepatic and biliary metabolism. It has been shown, at an oral dose of 5 × 10^10^ CFU/day for 12 weeks, to reduce total serum cholesterol in people with hypercholesterolemia; however, its effect is more subtle than the effect of hypocholesterolemic drugs [163]. There is evidence that the administration of this probiotic reduces visceral and subcutaneous fat, decreasing the BMI [164].

*Lactobacillus salivarius* has antimicrobial effects and improves bile tolerance. Its administration in humans with an oral capsule dose of 10^10^ CFU/day for 12 weeks promotes an increase in *Bacteroides*, *Prevotella*, and *Porphyromonas*, which is decreased by the excessive presence of the bacterial phylum *Firmicutes*, which characterizes the microbiota in people with obesity [165]. *Bifidobacterium animalis* subspecies *lactis* could have beneficial effects in reducing body adipose tissue, improving lipid profile, and increasing insulin sensitivity, as well as some effects in regulating redox balance and modulating satiety markers related to the metabolism of tryptophan, which includes the metabolism of serotonin and melatonin, an important regulator of the circadian sleep–wake rhythm [164]. There is clinical evidence that with the oral administration of this probiotic in a range of 10^10^ CFU/day, a reduction in abdominal, visceral, and subcutaneous fat is generated, and a reduction in anthropometric indicators of waist circumference is also evident in the BMI and taper index [166].

A clinical study was carried out with 45 patients with obesity, divided into three groups: single diet (low-carbohydrate and energy-reduced diet), prebiotics (30 g/day), and probiotics (a tablet containing *Bifidobacterium longum*, *Lactobacillus helveticus*, *Lactococcus lactis* and *Streptococcus thermophilus* per day). Age, sex, and BMI were used to categorize the three groups. All three groups significantly decreased weight, BMI, and waist circumference with *p* < 0.05. Only the prebiotic and probiotic groups showed a significant decrease in fat mass (*p* = 0.001) and a significant increase in muscle strength with *p* = 0.008 and 0.004, respectively. A significant reduction in fasting blood glucose was also demonstrated (*p* = 0.02).

It was concluded that the prescription of prebiotics and probiotics, together with lifestyle measures, seems interesting for the management of obesity, especially if it is sarcopenic, in addition to the improvement of metabolic parameters and psychiatric disorders related to obesity [167].

Table 6 shows the clinical effects of probiotics on obesity and related comorbidities, according to clinical trials and systematic reviews with meta-analyses of clinical trials found in the scientific literature.

### 3.7. Carotenoids

Carotenoids are a class of more than 750 natural pigments synthesized by plants, algae, and photosynthetic bacteria [170]. They are terpenoid compounds with 40 carbon atoms derived biosynthetically from two geranyl-geranyl-pyrophosphate units, so they are considered tetraterpenes [171]. These richly colored molecules are the source of many plants’ yellow, orange, and red colors. Fruits and vegetables provide most of the 40 to 50 carotenoids found in the human diet; the most common are α-Carotene, β-carotene, β-cryptoxanthin, lutein, zeaxanthin, and lycopene. The human body can transform α-Carotene, β-carotene, and β-cryptoxanthin, which are provitamin A carotenoids, to retinol. Lutein, zeaxanthin, and lycopene are non-provitamin A carotenoids because they cannot be converted to retinol [170].

One of the most widely studied carotenoids is astaxanthin (ASTX), widely known for its potent antioxidant capacity. However, it has other biological properties, such as anti-inflammatory, anti-aging, and anti-cancer effects. Recently, ASTX has been reported to inhibit the onset and development of fibrosis by regulating molecular signaling pathways, such as transforming growth factor β/ and small mother against decapentaplegic (TGF-β1/Smad), SIRT1, and nuclear factor. Kappa-B (NF-κB), microRNA, nuclear factor E2-related factor 2/antioxidant response element (Nrf 2/ARE), and ROS pathways [172], which are molecular signaling pathways that are altered in obesity, contribute to the development of comorbidities. Another study demonstrated that ASTX improves the activation of the Nrf-2/HO-1 signaling pathway, which promotes the process of mitophagy, inhibits oxidative stress and ferroptosis of cartilage endplate chondrocytes, and, finally, improves the degradation of the extracellular matrix, the calcification of cartilage endplate and endplate chondrocytes by activating apoptosis. Furthermore, ASTX inhibits the NF-κB activity induced by oxidative stimulation and can improve the inflammatory response [173]. ASTX could also protect the endplate of vertebral cartilage against oxidative stress and degeneration by activating the Nrf-2/HO-1 pathway [173]. However, dysfunction of this molecular pathway is related to the development of many pathologies, such as obesity.

Lycopene, another carotenoid belonging to the xanthophylls, has anti-obesity and anti-diabetic activities in different organs and/or tissues, including adipose tissue, liver, kidneys, pancreas, brain, ovaries, intestine, and eyes. The underlying mechanism may be attributed to its antioxidant and anti-inflammatory properties and its ability to regulate AGE/RAGE, JNK/MAPK, PI3K/Akt, SIRT1/FoxO1/PPARγ signaling pathways, and AchE activity [174]. Epidemiological research supports that lycopene consumption may reduce the risk of obesity and T2DM. The cis isomers of lycopene are more bioavailable and better absorbed than trans-lycopene and are mainly distributed in the liver and adipose tissue. Lycopene has a good safety margin and can be obtained by plant extraction, chemical synthesis, and microbial fermentation [174].

In a systematic review with a meta-analysis of randomized, placebo-controlled clinical trials from 2021 [175], the association between carotenoid intake and the presence of overweight and obesity was analyzed. Seven randomized controlled trials and eight observational studies containing 28,944 subjects and data on multiple carotenoid subgroups, including lycopene, astaxanthin, cryptoxanthin, α-carotene, and β-carotene, were included. In placebo-controlled clinical trials, the intervention lasted for a minimum of 20 days and a maximum of 16 weeks, with a dosage range of 1.2 to 60 mg/day. The meta-analysis found that serum carotenoid insufficiency was a risk factor for overweight and obesity (OR = 1.73, 95% CI [1.57, 1.91], *p* < 0.001). Furthermore, carotenoid supplementation was significantly associated with reduced body weight (SMD = −2.34 kg, 95% CI [−3.8, −0.87] kg, *p* < 0.001), decreased body mass index (BMI, SMD = −0.95 kg cm^2^, 95% CI [−1.88, −0.01] kg cm^2^, *p* < 0.001) and losses in waist circumference (WC, SMD = −1.84 cm, 95% CI [−3.14, −0.54] cm, *p* < 0.001). Therefore, it was concluded that carotenoids show promising effects in overweight or obese subjects. However, additional data from clinical trials are needed [175].

Table 7 shows the clinical effects of various carotenoids (carotenes and xanthophylls) on obesity and related comorbidities, according to clinical trials and systematic reviews with meta-analyses of clinical trials found in the scientific literature.

### 3.8. Curcumin

Curcumin, derived from the rhizome of *Curcuma longa*, is a natural pigment and the primary active component. It has been reported to have antioxidant, antiviral, antineoplastic, anti-inflammatory, and anti-obesity effects [178]. The mechanisms by which it can contribute to managing obesity are multiple since it can inhibit adipogenesis in both adipocyte differentiation and clonal expansion. It inhibits adipocyte differentiation by regulating the SREBP signaling pathway and increasing Rb phosphorylation and cyclin D1 expression [179]. Curcumin activates the phosphorylation of AMPK and consequently inhibits the expression of *PPARγ*, decreasing adipogenesis. Furthermore, curcumin inactivates fatty acid synthase by irreversible inhibition [180], which favors the prevention of hyperplasia of adipose tissue cells.

In humans, reports of the effect of curcumin on anthropometric parameters have been carried out in trials with patients with overweight or obesity combined with other pathologies. Although many studies have reported that curcumin reduces waist circumference [181,182] in patients with grade III obesity, no significant decrease in this variable or visceral fat was found [183]. In some clinical trials and meta-analyses, no significant reductions in BMI and weight have been found [181]; however, when performing the statistical analysis by groups, a decrease was found [182]. The results have been variable, and studies specifically designed to evaluate these parameters are required.

Although low doses of curcumin have not been related to adverse effects, some effects on anthropometric variables were obtained with high doses of this molecule [181,182]. High doses of curcumin, greater than 500 mg/day for 30 days, have been linked to nausea, dizziness, and liver damage [184]. Likewise, it has been reported that in populations with neoplastic risk, it can favor the appearance and development of malignant tumors through an oxidant effect more significant than the antioxidant [184,185]. In vitro studies report that it can inhibit the motility of healthy sperm, but clinical studies are required to determine its effect in healthy men after oral administration [184,186]. Caution is advised due to potential drug interactions, including with glibenclamide, as it inhibits the activity of CYP3A4 and may increase the risk of hypoglycemia [184].

Table 8 shows the clinical effects of curcumin on obesity and related comorbidities, according to clinical trials and systematic reviews with meta-analyses of clinical trials found in the scientific literature.

### 3.9. Silymarin/Silybin

Silymarin is the extract of *Silybum marianum*, commonly known as milk thistle, and comprises seven flavonolignans (including silibinin, isosilibinin, silicristin, isosylcristine and silidianin) and a flavonoid (taxifolin) [187]. Among these substances, silybin is the predominant one and has the most important biological effect. It constitutes approximately 70% of the total silymarin composition from two diastereoisomeric compounds: silybin A and silybin B [187]. Silymarin is a molecule with low bioavailability administered orally and poor solubility in water. This is due to its inefficient absorption in the intestine and a high metabolism of the first hepatic step after its absorption; both mechanisms reduce blood concentration and, consequently, arrive at the target organ [188]. However, this limitation has been effectively overcome with the introduction of complexes with phosphatidylcholine that have better absorption and new glycoconjugates of silibinin (gluco, manno, galacto, and lactoconjugates) that have high water solubility and strong antioxidant power [189]. No deaths or life-threatening adverse events have been reported in the medical and scientific literature [190].

Silymarin has been shown to inhibit the activity of the transcription factor NF-kB, reducing the expression of the genes encoding IL-1, IL-6, TNF-alpha, INF-gamma, and GMC-SF, which reduces inflammation [191]. Likewise, it inhibits the activation of MAPK induced by TNF-alpha, as well as the c-Jun N-terminal kinase, generating changes in the ratio of Bax/Bcl-2 proteins and inducing the release of cytochrome c from the mitochondria, activating initiator caspases 3 and 9 to activate the effector caspase cascade, and modulating IGF (insulin growth factor) signaling pathways, all of which regulate the correct apoptotic process [192]. Regarding its endocrine effects, silymarin partially activates estrogen receptors while presenting antiandrogenic activity in prostate cancer cells [193].

Additionally, silymarin can regulate various drug transporters by inhibiting membrane efflux proteins such as P-glycoprotein and the trypanosomal purine transporter TbAT1 [187]. Silymarin has also demonstrated antifibrotic effects, inhibiting the conversion of hepatic stellate cells into myofibroblasts, downregulating the expression of genes involved in fibrosis, such as procollagen III, *alpha-SMA*, and *TGF-beta* [187]. Silymarin also has a high antioxidant capacity, having the ability to neutralize free radicals, maintain the correct mitochondrial function, and activate the synthesis of protective molecules against oxidation and with reparative capacity, such as several heat shock proteins (HSPs), thioredoxin and sirtuins [194].

Silymarin has various beneficial metabolic effects, including serving as a PPAR-gamma agonist, improving the sensitivity of the insulin receptor type 1 substrate, increasing the activity of PI3K, Nrf2, and Akt, increasing the expression of *GLUT4* on the cell surface and inhibiting HMG-CoA reductase [195,196]. Silymarin has demonstrated choleretic activity by generating upregulation of the bile salt export pump [197,198].

Table 9 shows the clinical effects of silymarin/silybin on obesity and related comorbidities, according to clinical trials and systematic reviews with meta-analyses of clinical trials found in the scientific literature.

### 3.10. Hydroxycitric Acid

Hydroxycitric acid is an alpha-hydroxytribasic acid (1,2-dihydroxypropane-1,2,3-tricarboxylic acid) with two asymmetric centers, resulting in the formation of two pairs of diastereoisomers or four different isomers: acid (−) hydroxycitric acid (I), (+) hydroxycitric acid (II), (−) alo-hydroxycitric acid (III) and (+) alo-hydroxycitric acid (IV). The (−) hydroxycitric acid isomer (HCA) is found in the bark of the fruit of *Garcinia cambogia* (fam. *Clusiaceae*) [202,203].

Hydroxycitric acid is a competitive inhibitor of the enzyme ATP-citrate lyase, an enzyme that catalyzes the conversion of citrate and coenzyme A into oxaloacetate and acetyl coenzyme A in the cytosol. Acetyl CoA is necessary in the synthesis of fatty acids, cholesterol, and triglycerides, as well as in the synthesis of acetylcholine in the central nervous system. Inhibition of ATP-citrate lyase decreases acetyl CoA deposits, resulting in a reduction in malonyl CoA concentration, suppressing body fat accumulation through the activation of carnitine palmitoyl transferase I, an enzyme involved in the oxidation of fatty acids [203]. In some studies, it was observed that *G. cambogia* showed positive effects on the process of weight loss, lipogenesis, and reduction of appetite, percentage of body fat, triglycerides, cholesterol, and glucose levels. In contrast, in others, it had no effect [204].

In a prospective, non-randomized controlled intervention trial, 214 overweight or obese subjects were treated with *G. cambogia* and glucomannan (500 mg twice daily, each) for six months, evaluating weight, fat mass, visceral fat, rate basal metabolic rate, and blood profiles of lipids and glucose, comparing them with basal values. Some patients were carriers of the *PLIN4* polymorphisms -11482G > A-, fat mass and associated obesity (*FTO*) -rs9939609 A/T- and β-adrenergic receptor 3 (*ADRB3*) -Trp64Arg. The treatment produced weight loss, reducing fat mass, visceral fat, lipid profiles, and blood glucose while increasing basal metabolic rate. The results were independent of sex, age, or having hypertension, type 2 diabetes mellitus, or dyslipidemia and were attenuated in carriers of the *PLIN4*, *FTO*, and Trp64Arg polymorphisms [205]. However, multiple adverse effects have been reported [203,206]. A literature search study found 22 cases of liver injury caused by *G*. *cambogia* alone or in combination with green tea or Ashwagandha [207]. Patients taking *G*. *cambogia* were between 17 and 54 years old, and liver injury emerged between 13 and 223 days after onset. One patient died, one required a liver transplant, and 91% were hospitalized. The liver injury was hepatocellular with jaundice [207]. Table 10 shows the clinical effects of hydroxycitric acid *(G. cambogia*) on obesity and related comorbidities, according to clinical trials and systematic reviews with meta-analyses of clinical trials found in the scientific literature.

### 3.11. α-Lipoic Acid

α-Lipoic acid is a lipophilic thiol that, due to its chemical structure, is most easily identified by the name 6,8-dithio octanoic acid, although it is also often called 6,8-thioctic acid. The presence of two sulfhydryl radicals (-SH) in the structure of an eight-carbon fatty acid corresponds to its reduced form. Through an oxidation process, lipoic acid gives up two electrons and two protons (2e^−^ + 2H^+^) to form an intramolecular disulfide bridge, which makes it an effective antioxidant [211]. It is an organosulfur compound produced in plants, animals, and humans. Naturally, α-lipoic acid is found in the mitochondria and is used as a cofactor for pyruvate dehydrogenase (PDH) and α-ketoglutarate dehydrogenase complexes. Despite its diverse potential, the therapeutic efficacy of α-lipoic acid is relatively low due to its pharmacokinetic profile. Data suggest that α-lipoic acid has a short half-life and bioavailability (about 30%) caused by its hepatic degradation, reduced solubility, and instability in the stomach [212].

α-Lipoic acid treatment stimulates PI3K activity and insulin receptor substrate 1 (IRS-1) phosphorylation in 3T3-L1 adipocytes. Insulin receptor substrate phosphorylation involves the activation of further intracellular mediators and, subsequently, GLUT4 translocation; therefore, α-lipoic acid can be considered an insulin mimetic agent since insulin receptor binding and IRS-1 phosphorylation could subsequently lead to GLUT4 translocation, increasing glucose uptake [213]. α-Lipoic acid also decreases hypothalamic AMPK activity and causes rodent weight loss by reducing food intake and increasing energy expenditure [214]. Side effects of α-lipoic acid may include headache, tingling sensation, rash, or muscle cramps. There have been some reports in Japan of a rare case known as autoimmune insulin syndrome in people using α-lipoic acid [215]. This condition causes hypoglycemia and causes antibodies to attack the insulin produced by the body without having previous insulin treatment. The safety of α-lipoic acid in pregnant or lactating women, children, or people with liver or kidney disease is unknown. α-Lipoic acid can also have other adverse effects, such as hives, abdominal pain, nausea, diarrhea, and vomiting, as well as foul-smelling urine. However, these effects are dependent on the dose of lipoic acid and the administration [216].

In a preclinical study using a murine model, α-lipoic acid caused a notable reduction in the content of arachidonic acid, mainly in the phospholipid fraction, with a simultaneous decrease in the synthesis of proinflammatory mediators, that is, prostaglandin E2, leukotrienes B4 and C4, by decreasing the expression of *COX-2* and *LOX-5*. α-Lipoic acid also increased the level of antioxidants SOD2 and GSH and reduced the level of lipid peroxidation products, which improved the deterioration of the oxidative system in the tissue of the left ventricle of the heart. Therefore, α-lipoic acid has an important role in inflammation and the development of oxidative stress, decreasing the risk of obesity and cardiac dysfunction induced by a high-fat diet [217].

Clinical trials with α-lipoic acid treatment have shown a statistically significant short-term reductions in weight, BMI, body fat, and waist circumference, compared to placebo [218,219,220,221]. Table 11 shows the clinical effects of α-lipoic acid on obesity and related comorbidities, according to clinical trials and systematic reviews with meta-analyses of clinical trials found in the scientific literature.

## 4. Conclusions

Globally, obesity is one of the most widespread pathologies. In the same way, it demands a comprehensive and interdisciplinary strategy encompassing nutrition, physical education, anthropology, sociology, psychology, epidemiology, and medical treatments that can involve surgery and/or medication. The medications currently authorized for the treatment of obesity mainly generate a wide range of adverse effects, ranging from headaches to cardiac disorders that can cause the death of the individual. Alternatives of interest are phytopharmaceuticals and molecules of natural origin. In this review, we analyzed some natural molecules that have a promising future for obesity therapy and are primarily found in the usual human diet, which can contain hundreds to thousands of compounds with biological activity, depending on the typology and diversity of the diet. Due to their low concentrations in the conventional diet, these molecules often need to achieve therapeutic objectives or offer a minimal effect. Standardized presentations of these molecules may be an alternative for their administration. This review found that some molecules have few clinical research publications available.

All of the molecules included in this review have been evaluated in humans, demonstrating their clinical efficacy in vitro and in vivo. Some have results equal to or superior to those of currently authorized drugs, with the benefit of greater safety and lower incidence of adverse effects. However, some cause favorable modifications that reach statistical significance but lack clinical significance, making it essential to evaluate the synergies offered by different compounds to develop more complete and promising pharmaceutical formulations. The safety of some molecules, such as ellagic acid and resveratrol, places them in scientific interest since they have a wide therapeutic range and have not shown adverse effects. On the other hand, some molecules, such as hydroxycitric acid, must be carefully evaluated, as they carry the risk of serious adverse reactions. Therefore, some natural molecules deserve to be considered for further analysis as pharmacological treatment options, and others may be effective as a complementary therapy for the treatment of obesity, but more phase II and III studies are required to expand the evaluation of their safety and efficacy.

## Figures and Tables

**Table 1 ijms-25-02671-t001:** Clinical effects of epigallocatechin-3-gallate (EGCG) on obesity and associated comorbidities, from clinical trials and systematic reviews with meta-analyses.

Compound	Type of Study	Dose	Targeted Population	Observed Effect Δ	*p*	Adverse Effects	Reference
Epigallocatechin-3-gallate	Randomized, double-blind, placebo-controlled clinical trial in 30 patients.	150 mg/day orally, for 8 weeks.	Men and women who are overweight or obese.	Significant decrease in plasma triacylglycerides, systolic blood pressure, and diastolic blood pressure.	<0.05 in all parameters.	Unmentioned	Chatree et al., 2021 [83]
Epigallocatechin-3-gallate + α-glucosyl hesperidin	Clinical trial, randomized, placebo-controlled, double-blind, in parallel groups with 60 patients.	146 mg of EGCG + 178 mg of α-glucosyl hesperidin/day, orally, for 12 weeks.	Healthy Japanese men and women between 30 and 75 years old.	Reduction in BMI, triacylglycerides, body fat, visceral fat, and LDL-c/HDL-c ratio.	<0.05 in all parameters.	None	Yoshitomi et al., 2021 [82]

**Table 2 ijms-25-02671-t002:** Clinical effects of ellagic acid on obesity and associated comorbidities, from clinical trials and systematic reviews with meta-analyses.

Compound	Type of Study	Dose	Targeted Population	Observed Effect Δ	*p*	Adverse Effects	Reference
Ellagic acid	Double-blind randomized clinical trial with 32 patients.	1000 mg/day 12 weeks	Men and women with metabolic syndrome	Decrease in waist circumference H −2.7 cm M −3.8 cm	0.03 0.01	None	Hidalgo, et al., 2022 [92]
Ellagic acid	Double-blind randomized clinical trial with 32 patients.	3 mg/day 12 weeks	Overweight men and women	Decrease in waist circumference −0.7 cm	<0.01	None	Shiojima, et al., 2020 [93]
Ellagic acid	Double-blind randomized clinical trial with 150 patients.	50 mg/day 12 weeks	Overweight men	Decrease in waist circumference −1.5 cm	<0.01	None	Liu, et al., 2018 [94]

**Table 3 ijms-25-02671-t003:** Clinical effects of resveratrol on obesity and associated comorbidities, from clinical trials and systematic reviews with meta-analyses.

Compound	Type of Study	Dose	Targeted Population	Observed Effect Δ	*p*	Adverse Effects	Reference
Resveratrol	Systematic review and meta-analysis of 36 RCTs.	10 to 200 mg/day orally for 4–12 weeks.	Men and women with overweight or obesity, as well as comorbidities.	Decrease in body weight, BMI, fat mass, and waist circumference. Increase in lean mass. No significant changes in serum leptin and adiponectin levels.	0.03, 0.01, 0.03, 0.001 and <0.001, respectively.	None	Tabrizi et al., 2020 [111]
Resveratrol	Controlled, randomized, double-blind clinical trial with 71 participants.	1 g/day orally for 8 weeks.	Men and women with overweight and DM2.	It had no effect on hepatic steatosis or cardiovascular indices.	Not significant.	None	Ali-Sangouni et al., 2022 [112]
Resveratrol	Systematic review and meta-analysis of 19 RCTs with a total of 1151 patients.	>1 g/day orally for 4–12 weeks.	Men and women with DM2 and comorbidities such as overweight and obesity.	Reduction of fasting serum glucose and systolic and diastolic blood pressure. No significance in waist circumference, serum triacylglycerides, or HDL-c.	<0.00001, <0.00001, 0.95, 0.66 and 0.14, respectively.	None	Gu et al., 2022 [113]
Resveratrol	Systematic review and meta-analysis of 28 RCTs.	<500 mg/day orally for ≥3 months.	Men and women with obesity.	Reduction in body weight, BMI, and waist circumference. No effects on fat mass.	0.02, 0.02, 0.009 and 0.16, respectively.	None	Mousavi et al., 2019 [115]

**Table 4 ijms-25-02671-t004:** Clinical effects of berberine on obesity and associated comorbidities, from clinical trials and systematic reviews with meta-analyses.

Compound	Type of Study	Dose	Targeted Population	Observed Effect Δ	*p*	Adverse Effects	Reference
Berberine	Double-blind randomized clinical trial with 49 patients.	1100 mg/day for 8 weeks.	Men and women with impaired fasting glucose and overweight.	Decrease in fat mass −1.0 kg, visceral fat −93.2 g and waist circumference −1.42 cm.	<0.001 for the three variables	None	Rondanelli, et al., 2023 [133]
Berberine	Systematic review with meta-analysis of 9 clinical trials with a total of 378 individuals.	1000 to 1500 mg/day for 12–24 weeks.	Men and women who are overweight or obese in addition to having T2DM or metabolic syndrome.	Decrease in body weight. Decrease in BMI—0.29 kg/m^2^ and waist circumference −1.78cm	0.004 0.001	Not mentioned.	Xiong, et al., 2020 [130]
Berberine	Systematic review and meta-analysis of 12 clinical trials with a total of 849 individuals.	1000 to 1500 mg/day for 12–24 weeks.	Men and women who are overweight or obese in addition to having T2DM or metabolic syndrome.	No change in body weight or BMI. Decrease in waist/hip ratio—0.03	<0.001	Not mentioned.	Amini, et al., 2020 [132]

**Table 5 ijms-25-02671-t005:** Clinical effects of anthocyanins on obesity and associated comorbidities, from clinical trials and systematic reviews with meta-analyses.

Compound	Type of Study	Dose	Targeted Population	Observed Effect Δ	*p*	Adverse Effects	Reference
Anthocyanins	Randomized, placebo-controlled clinical trial with 55 participants.	200 g/day of açaí-juçara juice (293.6 mg) orally for 12 weeks.	Men and women who are overweight or obese.	Decrease in arterial stiffness (pulse wave speed) and peripheral vascular resistance. No changes in flow-mediated dilation (endothelial function).	0.002, 0.005 and >0.05, respectively.	None.	Arisi et al., 2023 [155]
Anthocyanins	Systematic review and meta-analysis of 11 RCTs with a total of 833 patients.	28.3–500 mg/day orally for 4–24 weeks.	Women and men who are overweight or obese.	Reduction in BMI and body weight. No significant changes in waist circumference.	0.002, 0.04, and >0.05, respectively.	Not mentioned.	Park et al., 2021 [156]

**Table 6 ijms-25-02671-t006:** Clinical effects of probiotics on obesity and associated comorbidities, from clinical trials and systematic reviews with meta-analyses.

Compound	Type of Study	Dose	Targeted Population	Observed Effect Δ	*p*	Adverse Effects	Reference
Probiotics	Clinical trial, randomized, controlled, and matched by age, sex, and BMI with 45 participants.	One tablet/day orally with *Bifidobacterium longum*, *Lactobacillus helveticus*, *Lactococcus lactis* and *Streptococcus thermophilus*, for 4 weeks.	Women and men with obesity.	Reduction in weight, BMI, and waist circumference. Decrease in fat mass. Increased muscle strength. Decrease in fasting blood glucose.	<0.05, 0.001, 0.004 and 0.02.	None	Ben Othman et al., 2023 [167]
Probiotics	Double-blind, randomized, placebo-controlled clinical trial with 81 participants.	One capsule/day orally with 13 million CFU/g of *Bacillus subtilis* (LMG P-32899) and *Bacillus coagulans* (LMG P-32921) for 12 weeks.	Overweight men and women between 18–45 years of age.	Reduction in body weight. No significant changes in BMI, waist circumference, blood pressure, or biomarkers.	0.027 and >0.05.	None	Danielsson et al., 2023 [168]
Probiotics	Randomized, double-blind, placebo-controlled clinical trial with 152 participants.	Oral capsules with *Lacticaseibacillus rhamnosus* HA-114 for 12 weeks.	Overweight adult men and women.	Significant decrease in plasma insulin, HOMA-IR, LDL cholesterol, and triacylglycerides. No significant changes in body weight and BMI.	<0.05 and >0.05.	None	Choi et al., 2023 [169]

**Table 7 ijms-25-02671-t007:** Clinical effects of carotenoids on obesity and associated comorbidities, from clinical trials and systematic reviews with meta-analyses.

Compound	Type of Study	Dose	Targeted Population	Observed Effect Δ	*p*	Adverse Effects	Reference
Carotenoids	Systematic review and meta-analysis of 7 RCTs and 8 observational studies, with 28,944 participants.	1.2–60 mg/day orally for 20 days to 16 weeks.	Men and women who are overweight or obese.	Insufficiency of serum carotenoids is a risk factor for overweight and obesity (OR = 1.73). Reduction in body weight, BMI, and waist circumference.	<0.001, <0.001, <0.001 and <0.001, respectively.	Not mentioned	Yao et al., 2021 [175]
Carotenoids	Systematic review and meta-analysis of 12 RCTs with a total of 380 participants.	Oral supplementation with astaxanthin in variable doses and periods.	Men and women with overweight, obesity, and/or DM2.	Significant reduction in blood malondialdehyde concentration and IL-6. Improvement of superoxide dismutase activity and reduction of serum isoprostane concentration.	<0.01, 0.02, <0.05 and >0.05.	Not mentioned	Ma et al., 2022 [176]
Carotenoids	Systematic review and meta-analysis of 7 RCTs with a total of 321 participants.	Oral supplementation of 0.16–20 mg/day for 8 weeks to 12 months.	Men and women with overweight or obesity and metabolic syndrome.	Significant reduction in LDL. No significant changes in BMI, fasting blood glucose, systolic and diastolic blood pressure, total cholesterol, HDL, or triacylglycerols.	<0.00001 and ≥0.05.	No serious adverse effects were reported.	Leung et al., 2022 [177]

**Table 8 ijms-25-02671-t008:** Clinical effects of curcumin on obesity and associated comorbidities, from clinical trials and systematic reviews with meta-analyses.

Compound	Type of Study	Dose	Targeted Population	Observed Effect Δ	*p*	Adverse Effects	Reference
Curcumin	Systematic review and meta-analysis of 8 clinical trials with 520 individuals.	70 to 3000 mg/day for 8 to 12 weeks	Overweight men and women with nonalcoholic fatty liver disease (NAFLD)	No change in body weight Decrease in BMI −0.34 kg/m^2^ and waist circumference −2.12 cm	<0.05 <0.01	Not mentioned.	Baziar et al., 2019 [181]
Curcumin	Review and meta-analysis of 14 SRMAs with 39 RCTs with 8111 individuals.	70 to 3000 mg/day orally for 8 to 12 weeks	Men and women with overweight or obesity, NAFLD, T2DM, PCOS, or MetS	Decreased body weight −0.59 kg BMI −0.24 kg/m^2^/ Waist circumference −1.32 cm		Not mentioned.	Unhapipatpong et al., 2023 [182]

**Table 9 ijms-25-02671-t009:** Clinical effects of silymarin/silybin on obesity and associated comorbidities, from clinical trials and systematic reviews with meta-analyses.

Compound	Type of Study	Dose	Targeted Population	Observed Effect Δ	*p*	Adverse Effects	Reference
Silymarin (*Silybum marianum*).	Systematic review and meta-analysis of 8 RCTs.	100–500 mg/day orally for 4–12 weeks.	Men and women with NAFLD (many of them overweight or obese).	Statistically significant reduction in BMI.	<0.05.	None.	Kalopitas et al., 2021 [199]
Silymarin (*Silybum marianum*).	Clinical, randomized, and controlled trial with 36 participants.	Two tablets/day orally of silymarin + vitamin E for 12 weeks.	Men and women with NAFLD (many of them overweight or obese).	Decrease in body weight and anthropometric parameters	<0.05 and >0.05.	None.	Aller et al., 2015 [200]
Silymarin (*Silybum marianum*).	Clinical, randomized, and controlled trial with 78 participants.	Two tablets/day of oral supplement with Curcuma longa, silymarin, guggul, chlorogenic acid, and inulin for 16 weeks.	Men and women with metabolic syndrome (many of them overweight or obese).	Significant body weight, BMI, waist circumference, fasting glucose and total cholesterol reduction.	<0.0001, 0.001, 0.0004, 0.014, 0.03 and >0.05, respectively.	None.	Patti et al., 2015 [201]

**Table 10 ijms-25-02671-t010:** Clinical effects of hydroxycitric acid on obesity and associated comorbidities, from clinical trials and systematic reviews with meta-analyses.

Compound	Type of Study	Dose	Targeted Population	Observed Effect Δ	*p*	Adverse Effects	Reference
Hydroxycitric acid (*G. cambogia*).	Randomized, placebo-controlled, double-blind, parallel-group clinical trial with 91 participants.	Two tablets a day 30 min before meals for oral administration for 14 weeks.	Caucasian men and women who are overweight or obese.	Weight loss. Reduction of fat mass, waist circumference, and hip circumference.	0.002 and <0.05.	There were no serious events reported.	Chong et al., 2014 [208]
Hydroxycitric acid (*G. cambogia*).	Clinical, randomized, and controlled trial with 86 participants.	Oral tablets at 2 g/day for 10 weeks.	Overweight men and women.	No significant effect on adipocytokines, non-HDL-c cholesterol, triacylglycerides, antioxidants, body weight, HDL-c, or total cholesterol.	>0.05.	Not mentioned.	Kim et al., 2011 [209]
Hydroxycitric acid (*G. cambogia*).	Randomized, double-blind, placebo-controlled clinical trial with 105 participants.	Polyherbal oral supplement in tablets with *G*. *cambogia* twice daily, for 12 weeks.	Men and women who are overweight or obese.	Significant change in the Body Composition Improvement Index. No significant changes in weight, BMI, and waist/hip ratio. Decrease in body fat.	0.012, >0.05 and 0.011.	Not mentioned.	Opala et al., 2006 [210]
Hydroxycitric acid (*G. cambogia*).	Case series in 1418 patients enrolled in the Drug-Induced Liver Injury Network (DILIN) from 2004 to 2018.	Oral supplementation of varying doses of *G*. *cambogia* alone or in combination with green tea or Ashwagandha.	Men and women between 17 and 54 years old.	22 cases of liver injury due to *G*. *cambogia* alone (*n* = 5) or in combination with green tea (*n* = 16) or Ashwagandha (*n* = 1), arising between 13 and 223 days after onset. Significant increase in aminotransferases.	<0.018	Liver injury. Hepatocellular injury with jaundice. Hospitalization. Requirement of liver transplant. Death.	Vuppalanchi et al., 2022 [207]

**Table 11 ijms-25-02671-t011:** Clinical effects of α-lipoic acid on obesity and associated comorbidities, from clinical trials and systematic reviews with meta-analyses.

Compound	Type of Study	Dose	Targeted Population	Observed Effect Δ	*p*	Adverse Effects	Reference
α-Lipoic acid	Clinical, controlled, and randomized trial with 92 participants	Oral administration for 8 weeks	Men and women with NAFLD and obesity	Reduction of alanine aminotransferase	0.012	Not mentioned	Tutunchi et al., 2023 [219]
α-Lipoic acid	Clinical, controlled, and randomized trial with 100 participants.	1200 mg/day orally for 8 weeks.	Men and women with obesity	Reduction in weight, BMI, body fat, and waist circumference	<0.05	Not mentioned	Mohammadshahi et al., 2022 [220]
α-Lipoic acid	Clinical, controlled, and randomized trial with 88 participants	600 mg/day orally for 16 weeks	Overweight women and men	Reduction in weight, waist circumference, and C-reactive protein. Maintenance of lost weight	<0.05	Not mentioned	Nasiri et al., 2021 [221]

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
