# Peer review of "Clinically Effective Molecules of Natural Origin for Obesity Prevention or Treatment"

_ijms, 2024, doi:10.3390/ijms25052671_

Round 1

Reviewer 1 Report

Comments and Suggestions for Authors

Review titled "Clinically effective molecules of natural origin for obesity prevention or treatment" presents results of current research that deals with the clinical effects of individual natural molecules united into a single topic of clinical effectiveness of natural molecules in obesity prevention and therapy.

All the references used when writing the review are of a recent date and were appropriately selected.

In the introduction, the authors state the basic characteristics of obesity, then the etiopathogenic mechanisms of obesity, after which they present data on clinical effectiveness of natural molecules from relevant research.

The work can represent a good starting point for researchers who are starting to investigate the field of clinical efficacy of secondary metabolites in obesity therapy and start planning their own research.

I think the review is very well. Below I list a couple of suggested improvements to the current text, in order to additionally improve the quality of the work:

1. After the introduction on obesity, it is necessary to state a short methodological part that explains the criteria by which the authors included the mentioned references in the review. I advise listing the key words that were used during the literature search. It would be good to present the selection of works and the inclusion/exclusion criteria graphically.

2. I suggest that the authors consider the possibility of adding a paragraph that would deal with the influence of a person's psychological state on obesity. In other words, an unhealthy diet affects biochemical changes in the brain, which then encourage the intake of food that supports obesity. Focus on proposed molecular mechanisms.

3. Did the authors find information in the literature on whether the type of pharmaceutical dosage form influenced the clinical effect of molecules of natural origin for obesity prevention or treatment?

4. During the revision, it is necessary to make some technical corrections to the text and check the language.

Comments on the Quality of English Language

Minor English corrections are needed. 

Author Response

Dear Reviewer

We appreciate the valuable recommendations and constructive comments made on our manuscript. Please see the attachment.

Reviewer 2 Report

Comments and Suggestions for Authors

General comments

 Overall this was a well-written comprehensive paper.  I applaud the authors for section 2.5 Gut microbiota, which is a hot topic as we all come to greater understanding how macronutrients affect the gut, and ultimately play a significant role in development of obesity. This maybe more significant that the energy balance theory held for years.

Specific Comments

1.    Section 2.3 Energy imbalance (lines 188-89)

The commentary that “the primary cause of obesity is excessive caloric intake compared to individuals' requirements” (lines 188-89) is outdated and not accepted.

Energy imbalance plays a role in the development of weight gain and becoming “overweight”. Obesity results from the changes that are induced by stressed adipose tissue to be continued over energy consumption and can also occur in non-overweight individuals. The energy imbalance theory is equal to the other mechanisms the authors have described in detail. I suggest changing that introductory sentence or the authors risk sounding outdated and behind in current understanding of obesity mechanism.

Appropriate wording would be “Excessive caloric intake compared to an individual’s requirements contributes to development of obesity.”

2.    Lines 200-202

Please double check your references as [23] discusses the changes that occur as a result of exercise, [35] The Global Status Report recommends 5 key areas for action to advance physical activity promotion. Neither of these references describe or address changes that occur as a result of a sedentary lifestyle.

This reference is also incorrect as [26] addresses maternal epigenetic regulation of gene expression and the fetus, and does not mention adipocyte numbers or predisposition of childhood/adolescent obesity and risks.

Author Response

(The authors gave the same response as above.)

Round 2

Reviewer 2 Report

Comments and Suggestions for Authors

Reference #135 does not appear to be cited in the text (Zhang at al.). Please either cite or remove it from the reference section. If I overlooked the citation, my apologies.

The previous suggestions and/or corrections have been addressed by the authors. I commend the authors for thier work on this in-depth comprehensive review paper.

Author Response

Dear reviewer:
I appreciate your valuable comments and words about our manuscript.
The aforementioned citation #135 is included in the text in its corresponding place (line 597).